# Myelodysplastic Syndromes with Isolated 20q Deletion: A New Clinical–Biological Entity?

**DOI:** 10.3390/jcm11092596

**Published:** 2022-05-05

**Authors:** Alessia Campagna, Daniela De Benedittis, Luana Fianchi, Emilia Scalzulli, Lorenzo Rizzo, Pasquale Niscola, Anna Lina Piccioni, Ambra Di Veroli, Stefano Mancini, Nicoletta Villivà, Tiziano Martini, Sara Mohamed, Ida Carmosino, Marianna Criscuolo, Susanna Fenu, Maria Antonietta Aloe Spiriti, Francesco Buccisano, Marco Mancini, Agostino Tafuri, Massimo Breccia, Antonella Poloni, Roberto Latagliata

**Affiliations:** 1Hematology, Sant’Andrea Hospital, Sapienza University, 00185 Rome, Italy; alessia.camp@gmail.com (A.C.); mariaantoniettaaloespiriti@gmail.com (M.A.A.S.); agostino.tafuri@ospedalesantandrea.it (A.T.); 2Hematology, Policlinico Umberto I, Sapienza University, 00185 Rome, Italy; danidb07@hotmail.it (D.D.B.); scalzulli@bce.uniroma1.it (E.S.); rizzo@bce.uniroma1.it (L.R.); mohamed@bce.uniroma1.it (S.M.); carmosino@bce.uniroma1.it (I.C.); mancini@bce.uniroma1.it (M.M.); breccia@bce.uniroma1.it (M.B.); 3Hematology, Policlinico Gemelli, University of Sacred Heart, 00168 Rome, Italy; luana.fianchi@policlinicogemelli.it (L.F.); marianna.criscuolo85@yahoo.it (M.C.); 4Hematology, Sant’Eugenio Hospital, 00144 Rome, Italy; pniscola@gmail.com; 5Hematology, San Giovanni Hospital, 00184 Rome, Italy; annalinapiccioni@virgilio.it (A.L.P.); sfenu@hsangiovanni.roma.it (S.F.); 6Hematology, Belcolle Hospital, 01100 Viterbo, Italy; ambra.diveroli@asl.vt.it; 7Hematology, San Camillo Hospital, 00152 Rome, Italy; smancini@scamilloforlanini.rm.it; 8Hematology, ASL RM1, 00193 Rome, Italy; nicoletta.villiva@aslrm1.it; 9AOU Ospedali Riuniti, Università Politecnica Marche, 60126 Ancona, Italy; t.martini@univpm.it (T.M.); a.poloni@univpm.it (A.P.); 10Hematology, University Tor Vergata, 00133 Rome, Italy; francesco.buccisano@uniroma2.it

**Keywords:** myelodysplastic syndromes, 20q deletion, erythropoietin, thrombocytopenia

## Abstract

Aims: To define the peculiar features of patients with the deletion of the chromosome 20 long arm (del20q), data from 69 patients with myelodysplastic syndromes (MDSs) and isolated del20q, followed by the Gruppo Romano-Laziale Sindromi Mielodisplastiche (GROM-L) and Ospedale Torrette of Ancona, were collected and compared with those of 502 MDS patients with normal karyotype (NK-MDS). Results: Compared to the NK-MDS group, patients with del20q at diagnosis were older (*p* = 0.020) and mainly male (*p* = 0.006). They also had a higher rate of bone marrow blast < 5% (*p* = 0.004), a higher proportion of low and int-1 risk according to IPSS score (*p* = 0.023), and lower median platelet (PLT) count (*p* < 0.001). To date, in the del20q cohort, 21 patients (30.4%) received no treatment, 42 (61.0%) were treated with erythropoiesis-stimulating agents (ESA), 3 (4.3%) with hypomethylating agents, and 3 (4.3%) with other treatments. Among 34 patients evaluable for response to ESA, 21 (61.7%) achieved stable erythroid response according to IWG 2006 criteria and 13 (38.2%) were resistant. Nine patients (13.0%) progressed to acute myeloid leukaemia (AML) after a median time from diagnosis of 28 months (IR 4.1–51.7). The median overall survival (OS) of the entire cohort was 60.6 months (95% CI 54.7–66.4). the 5-year cumulative OS was 55.9% (95% CI 40.6–71.2). Conclusion: According to our results, we hypothesize that MDSs with isolated del 20q may represent a distinct biological entity, with peculiar clinical and prognostic features. The physio-pathological mechanisms underlying the deletion of the chromosome 20 long arm are still unclear and warrant future molecular analysis.

## 1. Introduction

Myelodysplastic syndromes (MDSs) are a heterogeneous group of myeloid disorders characterized by defective bone marrow haematopoiesis that cause peripheral cytopenias, recurrent cytogenetic abnormalities, and a variable risk of progression to acute myeloid leukaemia (AML) [1].

Given the heterogeneity of the disease and the different clinical behavior in terms of leukemic evolution, there have been many efforts to identify patients with similar biological, clinical, and prognostic features [2]. These efforts led to the identification of two different entities among MDSs, characterized by the deletion of the long arm of chromosome 5 (del5q syndrome) or by the presence of ring sideroblasts and mutations in the spliceosome protein SFR3B1 (MDSs with ring sideroblasts).

Deletion of the long arm of chromosome 20 (del20q) has been observed in a small subset of MDS patients, both alone and with other cytogenetic abnormalities. In particular, del20q was reported in only 2% of the cohort for the validation of the IPSS and in 1.3% of the MDS included in the Groupe Francophone des Myélodysplasies (GFM) registry [3,4]. Descriptive analyses of clinical features in patients with isolated del20q are still limited due to its rarity. Isolated del20q has been associated with a low risk of progression to AML and prolonged survival compared to other MDS [3,4]. However, isolated del20q could be the hallmark of a relatively homogeneous subtype of MDS and thus deserves careful evaluation.

We collected data from all patients with isolated del20q MDSs treated in 12 hematologic centers to define their peculiar clinical and prognostic features in comparison to MDS patients of same prognostic group (IPSS) with normal karyotype (NK-MDS). The aim of this data collection is to evaluate the role of this karyotypic abnormality as the hallmark of a new possible subset in the MDS landscape.

## 2. Patients and Methods

### 2.1. Patients

We retrospectively collected data of all consecutive patients diagnosed as having MDS with isolated del20q between January 2002 and December 2016 in 12 haematologic centres of the Gruppo Romano-Laziale Sindromi Mielodisplastiche (GROM-L) and Ospedale Torrette of Ancona (six university hospitals and six community-based hospitals).

For enrolment in this study, the required criteria were a confirmed diagnosis of MDS according to WHO 2016 classification and the presence of isolated deletion of the long arm of chromosome 20. The presence of inherited bone marrow failure syndromes with del20q was an exclusion criterion.

Complete peripheral blood and bone marrow (BM) features were recorded for any eligible patients, together with follow-up data and the date of leukemic evolution. The International Prognostic Scoring System (IPSS) was used for the prognostic stratification of all patients. Characteristics of patients with isolated del20q were compared with those of 502 NK-MDS patients included in the GROM-L registry between January 2002 and December 2016.

Chromosome banding analysis was performed after 24 h on an unstimulated culture of BM cells using standard techniques. Karyotypes were described according to the International System for Human Cytogenetic Nomenclature criteria (2016). Cytogenetic analysis was performed at local specialized laboratories in each participating center.

### 2.2. Statistical Analysis

A descriptive analysis of the demographic and clinical characteristics of the patients was performed, including the median and interquartile range (IQR) for continuous variables, and the absolute and relative frequencies of categorical variables.

Overall survival (OS) was defined as the time from diagnosis to death from any cause or to the date of the last follow-up. Cumulative incidence of AML evolution was defined as the time from diagnosis to either the evolution into AML or the last follow-up.

Survival curves were estimated according to the Kaplan–Meier product limit method and were tested for significant differences using the log-rank test in univariate analysis and by means of the Cox regression model in multivariate analysis.

In all analyses, 95% confidence intervals (95% CI) were reported for the main summary statistics and all statistical comparisons were based on two-tailed tests accepting *p* ≤ 0.05 as statistically significant. All analyses were performed using the IBM SPSS statistics software.

## 3. Results

### 3.1. Baseline Characteristics and Comparison with NK-MDS

The study population included 69 patients with isolated del20q from a total number of 876 MDS patients with available karyotype, with a global incidence of 7.8%. The main clinical features of these patients are reported in Table 1 and compared with clinical features of NK-MDS patients. Unilineage marrow dysplasia was reported in 16 patients (23.2%) and was erythroid in 6 cases, granulocytic in 2 cases, and megakaryocytic in 8 cases; bilineage and trilineage dysplasia were reported in 27 (39.1%) and 26 (37.7%) of cases, respectively.

Compared with NK-MDS, patients with isolated del20q were older (*p* = 0.020) and mainly male (*p* = 0.006). They also had a higher rate of marrow blast count <5% (*p* = 0.004), a higher proportion of low/intermediate-1 risk according to IPSS (*p* = 0.023), and a lower median PLT count at diagnosis (*p* < 0.001). Limiting the comparison to patients with a blast count of <5%, patients with isolated del20q continued to have a lower median PLT count at diagnosis than the NK-MDS population [30/58 (51.7%) vs. 87/290 (30%) PLT count < 100 × 10^9^/L (*p* = 0.001)].

No significant differences were found between the two groups regarding leukocytes (WBC) count (*p* = 0.920), hemoglobin (Hb) (*p* = 0.849), and ferritin levels (*p* = 0.741).

Interestingly, nine patients with isolated del20q (13%) had a PLT count <100 × 10^9^/L with Hb values > 12 g/dL. These subjects were all males and all had marrow blasts <5%.

### 3.2. Treatment and Outcome of Patients with Isolated del 20q

At the last available follow-up, after a median time from diagnosis of 12.4 months (IQR 4.4–35.9), a watch-and-wait strategy was performed in 21 patients (30.4%), without starting any specific treatment.

Among the remaining 48 patients, 42 (61.0% of all patients with isolated del20q) were treated with erythropoiesis stimulating agents (ESA), 3 (4.3%) with hypomethylating agents, and 3 (4.3%) with other treatments (TNFα inhibitors, interferon, enrolment in a clinical trial).

Treatment with ESA started after a median time from diagnosis of 5.8 months (IQR 1.2–26.4): 27 patients (64.2%) received EPOα, 13 (31.0%) EPO-β, 1 (2.4%) darbopoietin, and 1 (2.4%) EPO-ζ. Median endogenous EPO levels at baseline of ESA treatment were 43.0 mU/L (IQR 20.7–50.6). Overall, 34 patients were fully evaluable for the response in accordance with the IWG 2006 criteria. Of these, 21 (61.8%) achieved a stable erythroid response (Hb increase >1.5 g/dL in 18 patients, reduction >50% of the baseline transfusion requirement in 3 patients), after a median time from ESA start of 3.2 months (IQR 2.1–5.6), while 13 (38.2%) were resistant. The median duration of response to ESA was 14.0 months (IQR 4.8–33.7).

Nine patients (13.0%) progressed to AML, with a median time to evolution of 28 months (IQR 4.1–51.7); 3-year and 5-year cumulative incidences were 10.2% (95% CI 1.6–18.8) and 20.0% (95% CI 4.9–35.1), respectively (Figure 1).

At the last follow-up, 30 patients (43.5%) died (13 for MDS/AML-related causes and 17 for unrelated causes), 14 (20.3%) were lost to follow-up, and 25 (36.2%) were alive.

The 3-year and 5-year OS of patients with isolated del20q were 62.9% (95% CI 49.2–76.6) and 55.9% (95% CI 40.6–71.2), respectively, with a median OS of 60.6 months (95% CI 54.7–66.4) (Figure 2).

The following features at diagnosis were evaluated as risk factors for cumulative incidence of AML evolution and OS in patients with isolated del20q: age (<75 years vs. ≥75 years), gender, Hb (<10 g/dL vs. ≥10 g/dL), PLT (<100 × 10^9^/L vs. ≥100 × 10^9^/L), WBC (<2.5 × 10^9^/L vs. ≥2.5 × 10^9^/L), marrow blasts (<5% vs. ≥5%), IPSS (low vs. intermediate-1), and transfusion need (no vs. yes).

At univariate analysis for cumulative incidence of AML evolution, a bone marrow blast percentage of ≥5% was the only feature significantly associated with higher risk of AML progression. The 3-year cumulative incidence of AML evolution was 50.8% (95% CI 32.0–69.6) for patients with ≥5% vs. 2.6% blasts (95% CI 0–7.5) and for patients with <5% blasts (*p* < 0.001) (Figure 3). All other variables were not statistically significant in terms of progression.

At univariate analysis for OS, factors associated with better survival were low IPSS (*p* < 0.001), age <75 years at diagnosis (*p* = 0.008) and Hb values >10 g/dL at diagnosis (*p* = 0.050); patients with age <75 years had a 3-year OS of 72.0% (95% CI 53.9–90.1) vs. 53.1% (95% CI 32.7–73.5) in patients aged ≥75 years. The remaining factors were not statistically significant.

On multivariate Cox regression analysis for OS, Hb values >10 g/dL [HR 0.39, 95% CI 0.16–0.94, *p* = 0.036] and age <75 years [HR 0.24, 95% CI 0.09–0.63, *p* = 0.004] retained an independent prognostic significance.

Comparing the cumulative incidence of AML evolution in patients with isolated del(20q) in the NK-MDS group, there was no statistically significant difference (*p* = 0.973) (Figure 1). On the other hand, MDS patients with isolated del20q had a 3-year OS significantly lower than the NK-MDS population (62.9% vs. 76.6%, *p* = 0.021) (Figure 2).

## 4. Discussion

We evaluated the clinical characteristics of 69 patients, which represent 3% of all MDS collected in the GROM-L registry from 2002 to 2016. As far as we know, this cohort of MDSs with isolated del20q is one of the widest in the current literature with the specific aim to compare this subtype with other MDSs; thus, we analyzed our results referring to the similar report of the GFM in 2011.

In the GFM analysis, patients with isolated del20q significantly differed from other MDS by higher reticulocyte count, lower platelet count, and lower mean percentage of marrow blasts [4]. These last two features were also confirmed in our cohort. Because of the low marrow blasts, most of the patients in our series belonged to low or intermediate-1 risk groups according to IPSS.

However, the most significant feature at diagnosis was a reduced PLT count, with values <100 × 10^9^/L in more than 50% of our patients, in accordance with GFM data. As a matter of fact, thrombocytopenia is highlighted as the commonest clinical distinctive feature in MDSs with del20q up to now.

As already reported by the GFM study, thrombocytopenia was associated with Hb values >12 g/dL in 13% of the subjects in our cohort, and in 7% of cases it represents the only cytopenia at diagnosis; these patients were all males and with marrow blasts <5%, thus raising the need of a careful differential diagnosis with idiopathic thrombocytopenic purpura (ITP). This point is strengthened by two different observations; first, it has already been pointed out in the literature that del20q is the most frequently encountered cytogenetic anomaly in patients with isolated thrombocytopenia at onset [5]. In addition, a previous analysis in which the clinical and morphological characteristics of patients with MDSs and isolated del20q were compared with those of patients diagnosed with ITP and critically revised shows a poor morphologic agreement among the three independent reviewers [6]. All these data highlight the importance of a cytogenetic analysis to search for the deletion of chromosome 20 in all elderly patients with isolated thrombocytopenia at diagnosis and in particular to those diagnosed as ITP, but not responsive to commonly effective treatments for this disease.

It is important to underline that the presence of del20q, as well as -Y and +8, is not considered to be MDS-defining in the absence of diagnostic morphologic features of MDSs; morphologic examination remains crucial in the differential diagnosis between ITP and MDS. It is worthy of note, to this respect, that some specific morphological features (hypogranulated and vacuolized neutrophils and neutrophil erythrophagocytosis) have already been reported in MDS subjects with del20q [7].

Moreover, patients with del20q were significantly older, with a median of 76 years at diagnosis, and were prevalently males compared to NK-MDS. These features were not reported in the GFM analysis, thus requesting further confirmations in larger series.

Regarding the treatment of patients with isolated del20q, about one third of them did not need any therapy and were managed with clinical monitoring only, while the vast majority received ESA, especially EPO-α. Erythroid response, according to the IWG criteria in these patients, was in line with other MDS subtypes, with more than 60% presenting a stable and durable response; thus, our data confirm the use of recombinant erythropoietin as the appropriate first therapeutic choice in these subjects.

The relatively good prognosis of patients with isolated del20q is also indicated by the low rate of AML evolution, similar to that reported in NK-MDS, and by the median OS of >60 months.

According to GFM data, the OS of these patients was influenced by anemia and thrombocytopenia. In our cohort, the presence of Hb < 10 g/dL was associated with a lower OS in both univariate and multivariate analysis; on the contrary, PLT count at diagnosis (even dividing patients by PLT values below 50 × 10^9^/L) was not significant.

Although the presence of isolated del20q is associated with a relatively favourable prognosis compared to patients with other karyotypic alterations, the OS remains worse compared to NK-MDS (*p* = 0.021). This observation may in part be explained by the significantly older age of patients with isolated del20q, but could also have a biological basis.

Large cohort studies have led to the identification of a putative commonly deleted region (CDR), located in the proximal area adjacent to the centromere (20q11.1–11.21) near the MAFB gene and in the sub-telomeric region (20q13.33) near the ADNP gene [8]. While the biological significance of del20q in the pathogenesis of MDSs is not yet clear, it is likely that some genes of CDR could be involved in this process. Several mechanisms were proposed, such as inactivation of tumor suppressor genes, allelic loss, and mutations with loss of function and haploinsufficiency (e.g., in del5q MDS patients) [9]. To date, two transcriptional regulators, L3MBTL1 and MYBL2, have been implicated in the pathogenesis of del20q-associated malignancies. The knockdown of L3MBTL1 causes an erythroid differentiation bias in human CD34+ cells, whereas MYBL2 inactivation promotes MDS development in aged mice [10,11]. Recently, Stoner et al. demonstrated that Hippo kinase STK4 is downregulated in del20q and that this inactivation in mice recapitulates clinical disease features [12]. In addition, ASXL1 chromosomal deletion was also detected in a significant number of patients with del20q, showing a negative prognostic impact on OS [13].

We are aware that the present retrospective evaluation has some important limitations. First of all, MDS diagnosis was not centrally revised, with possible discrepancies in the evaluation of dysplastic marrow features. Moreover, the vast majority of our patients were diagnosed many years ago, when mutational analyses were not performed. In addition, NGS techniques are still not used in the routine clinical activity. As a consequence, we do not have any mutational data to report in the present study, but we are planning a prospective phase to clarify this very important issue. Finally, isolated del20q has been reported also in other hematological disorders, both inherited and acquired, such as myeloproliferative neoplasms (MPN). Even if the presence of dysplastic marrow features and the absence of other signs typically reported in MPN (leuko-thrombocytosis, spleen enlargement) allowed us to discriminate our cases from MPN, we cannot exclude misdiagnosis at all.

In conclusion, MDSs with isolated del20q constitute a small but quite homogeneous subtype from a clinical and prognostic point of view, with a biological substrate likely different from other MDS. Differential diagnosis with ITP could be difficult in elderly patients with isolated thrombocytopenia, and cytogenetic or FISH analysis should be considered in some cases of refractory ITP.

## Figures and Tables

**Figure 1 jcm-11-02596-f001:**
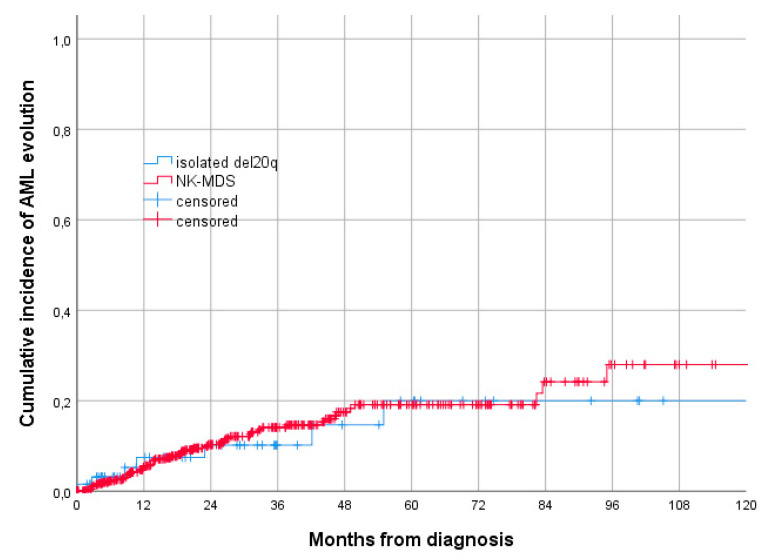
Cumulative incidence of AML evolution according to karyotype (3-year cumulative incidence of AML evolution 10.2% for del20q patients vs. 14.5% for patients with NK, *p* = 0.973, according to the log-rank test).

**Figure 2 jcm-11-02596-f002:**
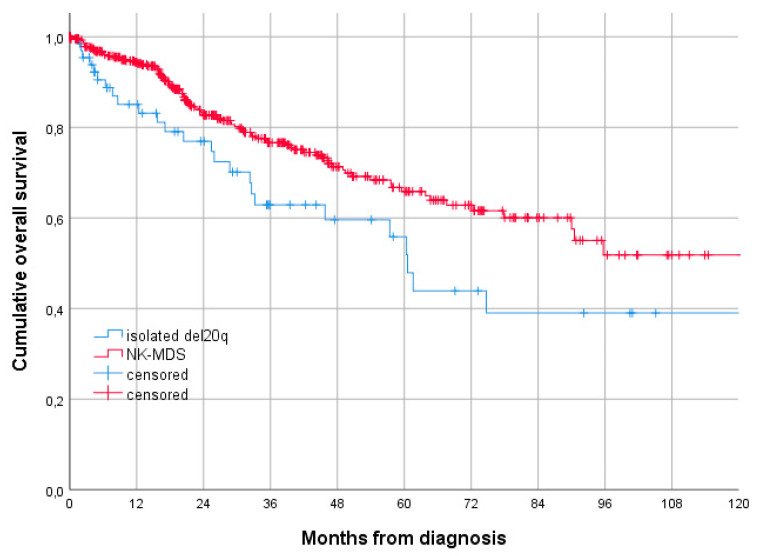
Cumulative overall survival according to karyotype (3-year OS 62.9% for del20q patients vs. 76.6% for patients with NK, *p* = 0.021, according to the log-rank test).

**Figure 3 jcm-11-02596-f003:**
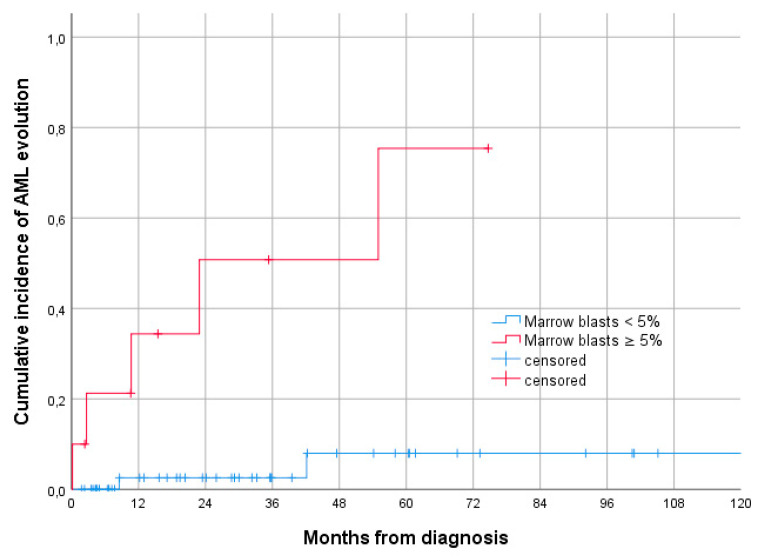
Cumulative incidence of AML evolution in isolated del20q MDS according to marrow blasts at diagnosis (3-year cumulative incidence of AML evolution 50.8% for patients with ≥5% vs. 2.6% blasts and for patients with <5% blasts, *p* < 0.001, according to the log-rank test).

**Table 1 jcm-11-02596-t001:** Patient features at diagnosis according to karyotype.

	MDS with del(20q)	MDS with NK	*p*
**M/F, n (%)**	50/19 (72.5/27.5)	276/226 (54.9/45.1)	0.006
**Median age, years (IQR)**	76.0 (68.9–81.5)	72.1 (63.2–73.3)	0.020
**WHO classification, n (%):**			
** MDS with unilineage dysplasia**	15 (21.7)	173 (34.5)	
** MDS with multilineage dysplasia**	40 (58.1)	138 (27.5)	
** MDS with sideroblasts**	3 (4.3)	24 (4.8)	<0.001
** MDS with excess of blasts 1**	6 (8.7)	76 (15.1)	
** MDS with excess of blasts 2**	4 (5.8)	66 (13.1)	
** MDS-Unclassifiable**	1 (1.4)	25 (5.0)	
**Marrow blasts, n° (%):**			
** <5%**	58 (84.0)	335 (66.7)	0.004
** ≥5%**	11 (16.0)	167 (33.3)	
**Median Hb, g/dL (IQR)**	10.2 (9.0–12.0)	10 (8.7–11.6)	0.849
**Median WBC, ×10^9^/L (IQR)**	3.7 (2.3–5.5)	3.6 (2.7–5.5)	0.920
**Median PLTS, ×10^9^/L (IQR)**	92 (51–133)	135 (75–230)	<0.001
**IPSS risk score, n (%):**			
** Low/Intermediate-1**	66 (95.6)	431 (85.8)	0.023
** Intermediate-2/High**	3 (4.4)	71 (14.2)	
**Median ferritin, mg/dL (IQR)**	213 (78–385)	193 (96–421)	0.741

## Data Availability

The data that support the findings of this study are available from the corresponding author upon reasonable request.

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
