# Peer review of "Myelodysplastic Syndromes with Isolated 20q Deletion: A New Clinical–Biological Entity?"

_jcm, 2022, doi:10.3390/jcm11092596_

Round 1
Reviewer 1 Report
I have the following concerns:
- Was a molecular analysis (mutations) for the MDS pts with del20q conducted? It would be of utmost importance to know which genes are mutated in this entity. Are there any data also in the literature? Please answer that to the results and the discussion respectively.
- You comment that the prognosis of MDS pts with del20q is favorable is in contradiction with the comparison of OS of del20q MDS pts of your cohort wuth the published NK-MDS. Do you think the prognosis of del 20q is rather intermediate, depending on the co-existence of various other mutations? Your explanation in the discussion is not enough.
- Did the MDS pts of your del20q cohort have specific morphological lesions, especially in megakaryocytes for the thrombopenic pts and/or other lineages and which were they? Please provide a relevant table describing the morphological lesions of your cohort that were indicative of MDS.
- Since many pts with del20q had thrombopenia and a very low number of blasts in the marrow, could we postulate that del20q is a subset of a hypoplastic MDS? Please answer to the discussion.
- The first paragraph of the discussion would better fit in the introduction, as it is.
- Few language errors: page 4 line 161: please omit 'In our study', page 4 line 195, 'in our cohort moreover', please remove again 'in our cohort', page 5, line 199 please remove the phrase 'As concern' and place 'Regarding' instead, page 5, line 233, 'it could be advisable' please replace the whole wrong phrase
Reviewer 2 Report
The authors report on isolated del (20q) from a cohort of 69 patients. The characteristics. This cohort was older, more likely male, lower blast count, and low-risk MDS by IPSS.
1. How often did del(20q) appear in complex cytogenetics of MDS? What were the total number of patients in the database (% of isolated del(20q) in MDS.
2. What were the results of the chromosome banding analysis as to where the deletion occurs on 20q?
3. Did any of the patients have Shwachman-Diamond syndrome which is associated with deletion of chromosome 20q ± myelodysplastic syndrome/acute myeloid leukemia?
4. Also on chromosome 20q is ASXL1, which is associated with myelodysplastic syndrome/clonal hematopoiesis of indeterminate potential. Was there any evidence of loss of heterozygosity for ASXL1?
5. Age range did not include anyone younger than 63 in the cohorts. Is there a referral or ascertainment bias among the six networked hospitals?
6. Were any of the MDS patients treated with hypomethylating agents?
Round 2
Reviewer 2 Report
There are two typos in line 107: erythroid, not erytroid; granulocytic, not granuoblastic.
More importantly, the figure legends are far too superficial. There needs to be more detail, including statistical analysis.
The text should state that no patients with inherited bone marrow failure syndromes were included -- if this is true -- but I wonder if there were some in the pre-NGS era.
Intermediate should only be used in the context of the IPSS-R. Does the criteria for these patients fit the IPSS-R criteria for intermediate.
